# Counterfactual Explanations for Neural Recommenders

Khanh Hiep Tran
MPI for Informatics, Germany
ktran@mpi-inf.mpg.de

Azin Ghazimatin
MPI for Informatics, Germany
aghazima@mpi-inf.mpg.de

Rishiraj Saha Roy
MPI for Informatics, Germany
rishiraj@mpi-inf.mpg.de

## ABSTRACT

Understanding why specific items are recommended to users can significantly increase their trust and satisfaction in the system. While neural recommenders have become the state-of-the-art in recent years, the complexity of deep models still makes the generation of tangible explanations for end users a challenging problem. Existing methods are usually based on attention distributions over a variety of features, which are still questionable regarding their suitability as explanations, and rather unwieldy to grasp for an end user. Counterfactual explanations based on a small set of the user's own actions have been shown to be an acceptable solution to the tangibility problem. However, current work on such counterfactuals cannot be readily applied to neural models. In this work, we propose ACCENT, the first general framework for finding counterfactual explanations for neural recommenders. It extends recently-proposed influence functions for identifying training points most relevant to a recommendation, from a single to a pair of items, while deducing a counterfactual set in an iterative process. We use ACCENT to generate counterfactual explanations for two popular neural models, Neural Collaborative Filtering (NCF) and Relational Collaborative Filtering (RCF), and demonstrate its feasibility on a sample of the popular MovieLens 100K dataset.

## ACM Reference Format:

Khanh Hiep Tran, Azin Ghazimatin, and Rishiraj Saha Roy. 2021. Counterfactual Explanations for Neural Recommenders. In *Proceedings of ACM SIGIR Conference (SIGIR'21)*. ACM, New York, NY, USA, Article 4, 5 pages. https://doi.org/10.1145/nnnnnnn.nnnnnnn

## 1 INTRODUCTION

**Motivation.** Recommender systems have become ubiquitous in today's online world, spanning e-commerce to news to social media. It is fairly well-accepted that high-quality explanations [5, 21, 25] for the recommended content can help improve users' satisfaction, while being actionable towards improving the underlying models [2, 9, 19, 22, 33, 34]. Typical methods explaining neural recommenders face certain concerns: (i) they often rely on the attention mechanism to find important words [26], reviews [4], or regions in images [6], which is still controversial [15, 27]; (ii) use connecting paths between users and items [1, 29, 31] that may not really be actionable and have privacy concerns; and, (iii) they use

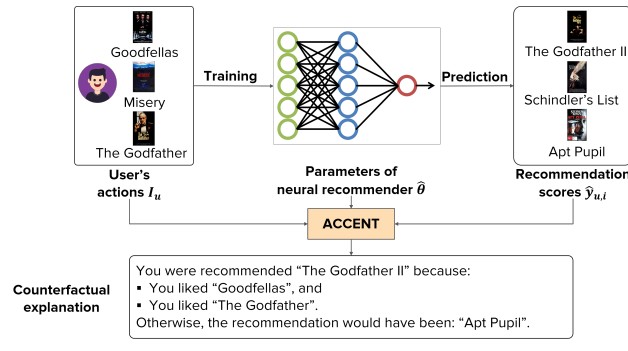

**Figure 1: Positioning ACCENT in a neural recommender setup. Taking the user's actions, parameters of the neural recommender, and predicted scores for items as input, ACCENT produces a concise counterfactual explanation.**

external item metadata such as reviews [18, 26, 28] or images [6, 28], that may not always be available.

In this context, it is reasonable to assume that in order to be tangible to end users, such explanations should relate to the user's own activity, and be scrutable, actionable, and concise [3, 10]. This paved the way to posit *counterfactual explanations* based on the user's own actions as a viable mechanism to address the tangibility concern [8, 16, 20, 23, 24]. A counterfactual explanation is a set of the user's own actions, that, when removed, produces a different recommendation (referred to as a *replacement item* in this text). In tandem with the huge body of work on explanations, recommender models themselves have continued to become increasingly complex. In recent years, neural recommender systems have become the de facto standard in the community, owing to their power of learning the sophisticated non-linear interplay between several factors [13, 30, 32]. However, this same complexity prevents us from generating counterfactual explanations with the same methodology that works well for graph-based recommenders (the PRINCE algorithm [8]).

**Approach.** To address this research gap, we present our method ACCENT (Action-based Counterfactual Explanations for Neural Recommenders for Tangibility), that extends the basic idea in PRINCE to neural recommenders. However, this necessitates tackling two basic challenges: (i) PRINCE relied on estimating *contribution scores* of a user's actions using Personalized PageRank for deriving counterfactual sets, something that does not carry over to arbitrary neural recommenders; and (ii) the graph-based *theoretical formalisms* that form the core of the PRINCE algorithm, and ensure its optimality, also do not readily extend to deep learning models. To overcome these obstacles, we adapt the recently proposed Fast Influence Analysis (FIA) [7] mechanism that sorts the user's actions based on their *approximate influence* on the prediction from the neural recommender. While such influence scores are a viable proxy for the contribution scores above, they cannot be directly used to produce

counterfactual sets. ACCENT extends the use of influence scores from single data points to *pairs of items*, the pair being the recommendation item and its replacement. This is a key step that enables producing counterfactual sets by iteratively closing the score gap between the original recommendation and a candidate replacement item from the original top-$k$ recommendations.

Figure 1 illustrates a typical counterfactual explanation output by ACCENT for the recommendation `The Godfather II`: had the user not watched movies `Goodfellas` and `The Godfather`, she would have been recommended `Apt Pupil` instead. More formally, given a user $u$, a list $I_u$ of items interacted by $u$ (her own actions), and a recommendation $rec$, ACCENT finds a counterfactual explanation $I_u^* \subseteq I_u$ whose removal from the training set results in a different recommendation $rec^*$. We apply ACCENT to explain predictions from two prominent neural recommenders, Neural Collaborative Filtering (NCF) [13] and Relational Collaborative Filtering (RCF) [30], and validate it on a subset of the MovieLens 100K dataset. All code and data are available at https://www.mpi-inf.mpg.de/impact/counterfactual-explanations-for-recommenders and https://github.com/hieptk/accent.

## 2 METHOD

### 2.1 Estimating parameters

Rooted in statistics, influence functions estimate how the model parameters change when a data point is upweighted by a small amount $\epsilon$ [11]. Using influence functions, Koh and Liang [17] proposed a method for estimating the impact of removing a data point from the training set (reducing its weight to 0) on the model parameters. To briefly describe this method, let $\hat{\theta}$ be the original model parameters, i.e., $\hat{\theta}$ is the minimizer of the empirical risk:

$$R(\theta) = \frac{1}{n} \sum_{i=1}^{n} L(z_i, \theta) \tag{1}$$

where $n$ is the number of training points and $L(z_i, \theta)$ is the loss of training point $z_i$ using parameters $\theta$. After upweighting training point $z$ by a small amount $\epsilon$, the new optimal parameters ($\hat{\theta}_{\epsilon,z}$) are approximated as follows:

$$\hat{\theta}_{\epsilon,z} = \arg\min_{\theta \in \Theta} \{R(\theta) + \epsilon L(z, \theta)\} \tag{2}$$

According to [17], the influence of upweighting training point $z$ by a small amount $\epsilon$ is given by:

$$\left. \frac{d\hat{\theta}_{\epsilon,z}}{d\epsilon} \right|_{\epsilon=0} = -H_{\hat{\theta}}^{-1} \nabla L(z, \hat{\theta}) \tag{3}$$

where $H_{\hat{\theta}}$ is the Hessian matrix computed as $H_{\hat{\theta}} = \frac{1}{n} \sum_{i=1}^{n} \nabla_\theta^2 L(z_i, \hat{\theta})$. Since the weight of each training point is $\frac{1}{n}$, removing a training point $z$ is equivalent to upweighting it by $-\frac{1}{n}$. We can estimate the new parameters $\hat{\theta}^{-z}$ after removing $z$ by setting $\epsilon = -\frac{1}{n}$ in (3):

$$\hat{\theta}^{-z} - \hat{\theta} = \frac{1}{n} H_{\hat{\theta}}^{-1} \nabla L(z, \hat{\theta}) \tag{4}$$

Note that, since neural models are not convex, we enforce the invertibility of $H_{\hat{\theta}}$ by adding a small damping term $\lambda$ to its diagonal.

A challenge while applying influence functions to neural models is that the set of parameters is usually huge. FIA [7] is a technique to reduce the computational cost of influence functions for recommender systems, and we will adapt this approach to our algorithm.

### 2.2 Computing influence on score gaps

Using FIA, we can estimate the new parameters $\hat{\theta}^{-z}$ when one training point (one of the user's actions $I_u$) is removed. Substituting $\hat{\theta}^{-z}$ into the recommender model, we can estimate the new predicted score for any item. We denote the preference score of user $u$ for item $i$ before and after removing point $z$ as $\hat{y}_{u,i}$ and $\hat{y}_{u,i}^{-z}$, respectively. The influence of $z$ on $\hat{y}_{u,i}$ is defined as:

$$I(z, \hat{y}_{u,i}) = \hat{y}_{u,i} - \hat{y}_{u,i}^{-z} \tag{5}$$

We can then estimate the influence of a training point $z$ on the *score gap* between two items $i$ and $j$ as:

$$I(z, \hat{y}_{u,i} - \hat{y}_{u,j}) = (\hat{y}_{u,i} - \hat{y}_{u,j}) - (\hat{y}_{u,i}^{-z} - \hat{y}_{u,j}^{-z})$$
$$= (\hat{y}_{u,i} - \hat{y}_{u,i}^{-z}) - (\hat{y}_{u,j} - \hat{y}_{u,j}^{-z}) = I(z, \hat{y}_{u,i}) - I(z, \hat{y}_{u,j}) \tag{6}$$

For a set of points $Z = \{z_1, z_2, ..., z_m\}$, we approximate the influence of removing this set by:

$$I(Z, \hat{y}_{u,i} - \hat{y}_{u,j}) = \sum_{k=1}^{m} I(z_k, \hat{y}_{u,i} - \hat{y}_{u,j}) \tag{7}$$

With the increase in the size of the set $Z$, the accuracy of the above estimation deteriorates. However, since counterfactual sets are usually very small, this approximation is still valid.

### 2.3 Filling the gap

To replace the recommendation $rec$ with $rec^*$, we need to find a counterfactual explanation set $Z \subseteq I_u$ whose removal results in:

$$\hat{y}_{u,rec}^{-Z} - \hat{y}_{u,rec^*}^{-Z} < 0$$
$$\Leftrightarrow \hat{y}_{u,rec} - \hat{y}_{u,rec^*} - \hat{y}_{u,rec}^{-Z} + \hat{y}_{u,rec^*}^{-Z} > \hat{y}_{u,rec} - \hat{y}_{u,rec^*}$$
$$\Leftrightarrow I(Z, \hat{y}_{u,rec} - \hat{y}_{u,rec^*}) > \hat{y}_{u,rec} - \hat{y}_{u,rec^*}$$
$$\Leftrightarrow \sum_{k=1}^{m} I(z_k, \hat{y}_{u,rec} - \hat{y}_{u,rec^*}) > \hat{y}_{u,rec} - \hat{y}_{u,rec^*} \tag{8}$$

Therefore, the optimal way to replace $rec$ with $rec^*$ is to add training points $z_k$ to $Z$ in the order of decreasing $I(z_k, \hat{y}_{u,rec} - \hat{y}_{u,rec^*})$ until (8) is satisfied. To find the smallest counterfactual explanation $I_u^*$, we try every replacement item from a set of candidates $I_{rep}$. In principle, $I_{rep}$ could span the complete set of items $I$, but a practical choice is the original set of top-$k$ recommendations. Good models usually diversify their top-$k$ while preserving relevance, and choosing the replacement from this slate ensures that $rec^*$ is neither trivially similar to $rec$ nor irrelevant to $u$. Finally, this smallest set of actions $I_u^*$ is returned to $u$ as a tangible explanation for $rec$. **Algorithm 1** contains a precise formulation of ACCENT. ACCENT's time complexity is $O(|I_{rep}| \times |I_u| \times \log |I_u|) + O(|I_{rep}| \times |I_u|)$ calls of FIA. Since ACCENT only requires access to gradients and the Hessian matrix, it is applicable to a large class of neural recommenders.

## 3 EXPERIMENTAL SETUP

### 3.1 Recommender models

We apply ACCENT on NCF [13], one of the first neural recommenders, and RCF [30], a more recent choice. NCF [13] consists of a

**Algorithm 1:** ACCENT

> **Input:** user $u$, recommendation item $rec$,
> items interacted with by user $I_u$, candidate replacement items $I_{rep}$
> **Output:** smallest counterfactual set $I_u^*$, replacement $rec^*$
>
> $I_u^*, rec^* \leftarrow I_u, -1$
> **for** $i \in I_{rep}$ **do**
>      **for** $z \in I_u$ **do**           // Compute influence on gap
>          **compute** $I(z, \hat{y}_{u,rec})$ and $I(z, \hat{y}_{u,i})$
>          $I(z, \hat{y}_{u,rec} - \hat{y}_{u,i}) \leftarrow I(z, \hat{y}_{u,rec}) - I(z, \hat{y}_{u,i})$
>      **end**
>      $gap, I_u^i \leftarrow \hat{y}_{u,rec} - \hat{y}_{u,i}, \emptyset$
>      **sort** $I_u$ by decreasing $I(z, \hat{y}_{u,rec} - \hat{y}_{u,i})$
>      **for** $z \in sorted(I_u)$ **do**
>          **if** $gap < 0$ **or** $I(z, \hat{y}_{u,rec} - \hat{y}_{u,i}) \leq 0$ **then**
>              // Gap is filled or impossible
>              **break**
>          **end**
>          $gap \leftarrow gap - I(z, \hat{y}_{u,rec} - \hat{y}_{u,i})$      // Update gap
>          $I_u^i \leftarrow I_u^i \cup \{z\}$             // and result set
>      **end**
>      **if** $gap < 0$ **and** $|I_u^i| < |I_u^*|$ **then**      // New smallest set
>          $I_u^*, rec^* \leftarrow I_u^i, i$
>      **end**
> **end**
> **return** $I_u^*, rec^*$

generalized matrix factorization layer (where the user and the item embeddings are element-wise multiplied), and a multilayer perceptron that takes these user and item embeddings as input. These two parts are then combined to predict the final recommendation score. RCF uses auxiliary information to incorporate item-item relations into the model. It computes target-aware embeddings that capture information about the user, her interactions, and their relationships with target items (recommendation candidates) using a two-layer attention scheme. The recommendation score of the user and the target item are computed from these target-aware embeddings.

### 3.2 Dataset

We use the popular MovieLens 100K dataset [12], which contains $100k$ ratings on a $1 - 5$ scale by 943 users on 1682 movies. Input of this form can be directly fed into NCF. On the other hand, to conform to the implicit feedback setting in RCF, we binarized ratings to a positive label if it is 3 or above, and a negative label otherwise. We removed all users with $< 10$ positive ratings or $< 10$ negative ratings so that the profiles are big and balanced enough for learning discriminative user models. This pruning results in 452 users, 1654 movies, and 61054 interactions in our dataset. For item-item relations in RCF, we used the auxiliary data provided in [30], which contains four relation types and 97209 relation pairs.

### 3.3 Baselines

We compare ACCENT against four baseline algorithms. Two baselines are based on the attention weights in RCF and are not applicable to NCF, while the other two algorithms are based on FIA scores and can be used for both recommender models.

*3.3.1 Attention-based algorithms.* Attention weights in RCF can be used to produce explanations [30]. An item's attention weight

shows how much it affects the prediction: thus, to find a counterfactual explanation for $rec$, we can sort all items in $I_u$ by decreasing attention weights. We then add these items one by one to $I_u^*$ until the predicted recommendation is changed. Here, we assume removing a few training points does not change the model significantly, so all parameters remain fixed. We refer to this as **pure attention**. We adapt pure attention to a smarter **attention** baseline, where an item is added to $I_u^*$ only if removing it reduces the score gap between $\hat{y}_{u,rec}$ and the second-ranked item's score. The underlying intuition is to avoid adding potentially irrelevant items to the explanation. The score gap is again estimated using fixed parameters.

*3.3.2 FIA-based algorithms.* Here, we test the direct applicability of FIA to produce counterfactual explanations. We simply sort items in $I_u$ by $I(z, \hat{y}_{u,rec})$ and add items one by one to $I_u^*$ until $rec$ is displaced (**pure FIA**). As with attention, we improve pure FIA to keep only the interactions that reduce the score gap between $\hat{y}_{u,rec}$ and the score of the second-ranked item (strategy denoted by **FIA**).

### 3.4 Initialization

For FIA on NCF, we used the implementation in [7]. We used a batch size 1246 as this implementation requires this value to be a factor of the dataset size (61054). All other hyperparameters were kept the same. For RCF, we set the dropout rate to 0 to minimize randomness during retraining. We replaced the ReLU activation with GELU [14] to avoid problems with non-differentiability [17]. To guarantee FIA's effectiveness, we made sure that each interaction corresponds to one training point (that was fifty in the original model). For this, we paired each liked item $i^+$ by user $u$ with one of her disliked items $i^-$, and added triples $(u, i^+, i^-)$ to the training set. In particular, for each $i^+$, we selected an $i^-$ that shares the highest number of relations with $i^+$. By doing this principled negative sampling, the RCF model can still discriminate between positive and negative items effectively, despite having only one negative item for each positive. For FIA on RCF, we added damping term $\lambda = 0.01$ to the Hessian matrix and used our own implementation.

## 4 RESULTS AND INSIGHTS

### 4.1 Evaluation protocol

For each of the 452 users in our dataset, we find an explanation $I_u^*$ for their recommendation $rec$, and a replacement $rec^*$ from $I_{rep}$, where $I_{rep}$ is the original top-$k$ ($k = 5, 10, 20$). We then retrain the models without $I_u^*$ and verify if $rec^*$ replaces $rec$. This is done for both recommender models (NCF and RCF) and each of the explanation algorithms (ACCENT, pure attention, attention, pure FIA and FIA, as applicable). The percentages of actual replacements (CF percentage) and the average sizes of the counterfactual sets (CF set size) for ACCENT and the baselines are reported in Table 1. Ideally, an algorithm should have a high CF percentage and a small CF set size. To give a qualitative feel of the explanations generated by ACCENT, we provide some anecdotal examples in Table 2 (baselines had larger CF sets). To compare the counterfactual effect (CF percentages) between two methods, we used the McNemar's test for paired binomial data, since each explanation is either actually counterfactual or not (binary). For CF set sizes, we used the one-sided paired $t$-test. The significance level for all tests was set to 0.05.

| Candidate top-$k$ set of replacement items | | $k = 5$ | | $k = 10$ | | $k = 20$ | |
|---|---|---|---|---|---|---|---|
| Recommender model | Explanation model | CF percentage | CF set size | CF percentage | CF set size | CF percentage | CF set size |
| NCF [13] | **Pure FIA** [7] | 54.20 | 9.08 | 56.19 | 9.46 | 55.75 | 9.50 |
| | **FIA** [7] | 55.97 | 7.98 | 56.19 | 7.80 | 55.75 | 7.84 |
| | **ACCENT (Proposed)** | **57.30** | **4.73*** | **57.74** | **4.69*** | **57.08** | **4.62*** |
| RCF [30] | **Pure Attention** [30] | 73.01 | 9.36 | 73.45 | 7.94 | 74.34 | 7.75 |
| | **Attention** [30] | 76.99 | 3.55 | 76.99 | 3.53 | 76.99 | 3.51 |
| | **Pure FIA** [7] | 80.75 | 4.85 | 81.19 | 4.62 | 81.86 | 4.72 |
| | **FIA** [7] | 81.64 | 4.15 | 81.86 | 4.10 | 81.86 | 4.10 |
| | **ACCENT (Proposed)** | **81.86**† | **2.83***† | **82.08**† | **2.75***† | **82.08**† | **2.74***† |

Best values in each column are in **bold**. * and † denote statistical significance of ACCENT over FIA and Attention, respectively.

**Table 1: Performance comparison of ACCENT with baselines on our sample of the MovieLens 100K benchmark.**

| Recommendation | ACCENT Explanation | Replacement |
|---|---|---|
| The Silence Of The Lambs | Contact
Fargo | Donnie Brasco |
| Titanic | True Romance
The Basketball Diaries | East Of Eden |
| The Devil's Advocate | Speed
Eraser
It's A Wonderful Life | My Fair Lady |

**Table 2: Counterfactual sets generated by ACCENT.**

## 4.2 Key findings

**ACCENT is effective.** For both models, ACCENT produced the best overall results for producing counterfactual explanations. Results are statistically significant for CF percentages over attention baselines, and for CF set sizes over both attention and FIA methods (marked with asterisks and daggers in Table 1).

**Attention is not explanation.** The two algorithms using attention performed the worst. Their CF percentage is at least 5% lower than ACCENT and their average CF set sizes are between 1.3 to 3.3 times bigger than ACCENT. This shows that using attention is not really helpful in finding concise counterfactual explanations.

**FIA is not enough.** The two algorithms that directly use FIA to rank interactions produced very big explanations. The average size of FIA's explanations is 1.5 to 1.7 times bigger than that of ACCENT (about twice as big for pure FIA with NCF, the context in which FIA was originally proposed). This provides evidence that to replace *rec*, the influence on the score gap is more important than the influence on the score of *rec* alone.

**Considering the score gap is essential.** The two pure algorithms that do not consider the score gap between *rec* and the replacement while expanding the explanation, performed worse than their smarter versions that do take this gap reduction into account. The difference can be as large as 4% in CF percentages, with up to 2.6 times bigger CF sets (pure attention, $k = 5$).

**Influence estimation for sets is adequate.** Our approximation of FIA for sets of items is actually quite close to the true influence. For RCF, the RMSE between the approximated influence and the true influence is 1.17 over different values of $k$, which is small compared to the standard deviation of the true influence ($\simeq 3.7$). For NCF, this RMSE is 0.36 while the true influence has a standard deviation of 0.34, implying that estimation accuracy is lower for this model: this in turn results in a lower CF percentage.

**Explanations get smaller as $k$ grows.** The performance of ACCENT is stable across different $k$ (5, 10, 20), varying less than 1%.

The average CF set size slightly decreases as $k$ increases, because we have more options to replace *rec* with. Interestingly, a similar effect was observed in the graph-based setup in PRINCE [8].

## 4.3 Analysis

**Pairwise vis-à-vis one-versus-all.** In our main algorithm, instead of fixing one replacement item at a time (pairwise), we can have a different approach that does not need a fixed replacement. In particular, at each step, we can reduce the gap between *rec* and the second-ranked item at the time, leaving this second-ranked item to change freely during the process (one-versus-all). Through experiments, we found that this approach can slightly improve the counterfactual percentage of ACCENT by 0.22% on RCF but at the cost of bigger explanations.

**Error analysis.** Despite being effective in estimating influence, FIA is still an approximation. In particular, it assumes that only a few parameters are affected by the removal of a data point (corresponding user and item embeddings). This assumption can sometimes lead to errors in practice. In NCF, we observed a large discrepancy between the estimated influence and the actual score drops, despite their strong correlation ($\rho = 0.77$). This explains ACCENT's relatively low CF percentages in NCF ($\simeq 57\%$). It would thus be desirable to update more parameters when one action is removed: for example, in RCF, we can also consider relation type and relation value embeddings. However, this could substantially increase the computational cost. Another source of error is that the influence of a set of items is sometimes overestimated. This can cause ACCENT to stop prematurely when the cumulative influence is not enough to swap two items. To mitigate this, we can retrain the model after ACCENT stops, to verify whether the result is actually counterfactual. If not, ACCENT can resume and add more actions.

## 5 CONCLUSION

We described ACCENT, a mechanism for generating counterfactual explanations for neural recommenders. ACCENT extends ideas from the PRINCE algorithm to the neural setup, while using and adapting influence values from FIA for the pairwise contribution scores that were a core component of PRINCE but non-trivial to obtain in deep models. We demonstrated ACCENT's effectiveness over attention and FIA baselines with the underlying recommender being NCF or RCF, but it is applicable to a much broader class of models: the only requirements are access to gradients and the Hessian.

**Acknowledgements.** This work was supported by the ERC Synergy Grant 610150 (imPACT).

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
