# OpenReview forum: "Counterfactual Explanations for Neural Recommenders"
_ACM.org/SIGIR/Badging_

### Official Review · ~Timo_Breuer1 · 2021-08-04
**Approved**

**Comment:**

Dear Khanh, Azin, and Rishiraj,

Thank you for addressing the open issues so thoroughly!

We are happy to award you the following badges:
* Artifacts Evaluated – Functional
* Artifacts Evaluated – Reusable and Available

Many thanks and best regards,
Timo

**Awarded Badges:**

["Artifacts Evaluated – Functional", "Artifacts Evaluated – Reusable and Available"]

---

### Official Review · Program_Chairs · 2021-08-11
**Functional, Reusable, and Available Badges**

**Comment:**

According to the discussion you had with the reviewers and to the final reviews, your artifact is ready for badging.

We are happy to award you the following badges:
* Artifacts Evaluated – Functional
* Artifacts Evaluated – Reusable and Available

To have your artifact included in the ACM DL we need to received a single zip file containing it.

Could you, please, send it to us at the following email address:

aec_sigir@acm.org

at your earliest convenience?

**Awarded Badges:**

["Artifacts Evaluated – Functional", "Artifacts Evaluated – Reusable and Available"]

---

### Public Comment · ~Timo_Breuer1 · 2021-07-30
**Necessary actions**

Dear Khanh, Azin, and Rishiraj,

Many thanks for providing the artifacts of your SIGIR publication! We really appreciate how much effort you have put into preparing it for future re-use!

Overall, we do not see any issues awarding the "Functional" badge, but we have some minor concerns, tagged with (-), awarding the "Reusable and Available" badge. Some more effort should still be made regarding the documentation, code structure, explicitness of which commit was used, and the data archives (details below). We do not expect to take it more than 2-3 hours.

I've tested everything on a fresh Ubuntu 20.04 LTS without any GPUs installed.

Best regards,
Timo

### Functional

"The artifacts associated with the research are found to be"

- "documented"

    (+) The repository contains a detailed README file.

    (+) The classes and functions are properly documented and follow Google's formatting style. "accent/NCF/src/scripts/hessians.py" is an example of good code documentation.

- "consistent"

    (+) The pruned data complies with the descriptions provided in the paper.

    (+) The code repository includes one directory for each of the two analyzed neural recommenders and the corresponding scripts for training/evaluation/etc.

- "complete"

    (+) All scripts for training, conducting and evaluating the experiments are included. The original dataset (MovieLens 100K) is publicly available, scripts for the data pruning are provided and the intermediate results are also available via an external file hosting service.

- "exercisable"

    (+) The scripts run without errors and produce results.

    (+) The required packages and versions are documented in requirements.txt

- "and include appropriate evidence of verification and validation."

    (+) I do not see any issues here.

- "the artifact should be made available in an online repository; the artifact corresponds to what is mentioned in the corresponding paper"

    (+) The code is hosted on GitHub and is released with GPL-3.0 License. The names of the directories, scripts, functions, etc. comply with those names in the paper. The data is hosted on Mega and is retrievable at the time of review.


### Reusable and Available

- "The starting point of the guide is a freshly installed OS."

    (+) The authors explicitly mention that Ubuntu 20.04 LTS has to be used.

- "The README of the code repository contains a step-by-step deployment guide."

    (+) The instructions provided in the README were easy to follow and work as expected.

- "The code itself need to be properly commented and documented"

    (-) The classes and functions are properly documented, but some of the scripts lack code documentation. It would be nice to have a broad overview of the Python modules in RCF/ and NCF/.

    (-) The subdirectories RCF and NCF are not consistent. Maybe it is not feasible due to the different nature of the two approaches, but I see some code fragments (functions) repeated for both approaches (e.g. in accent.py, fia.py). In order to make code maintenance and future re-uses easier, it would be helpful to combine shared functions in single  modules, introduce class inheritance, and to harmonize the directory structures.

- Packaging, successful compilation and execution

    (+) In addition to the repository, the authors package the entire experimental setup in a virtual machine. The required Python packages are pre-installed and the image also contains the pruned data - everything works as advertised!

    (-) It is not clear to me which commit was used for the experiments (git tags could be used). I assume, it was the latest commit (the README points to a zipped archive of it). This could have been made more explicit. If the ACCENT framework should be extended in the future, release versioning might also be helpful.

- "For datasets, a description of their “schema” is also expected and an explanation of the intended way to process them"

    (+) The README provides a good description of the data and the corresponding schema. This information could also added be as an additional README file to the directories of the data.

    (+) The scripts load the (intermediate) data ouputs of previous steps without any issues.

    (+) Data provenance is transparent. The README provides a good description of the data schema.

    (-) The data (intermediate results, pretrained models, experimental results, virtual machine) is hosted on Mega. For long-term reproducibility, I'd rather recommend to use a single citable repository, e.g. hosted on Zenodo or figshare.

- "Check that the content of the artifact corresponds to its description in the paper"

    (+) The compare.py script outputs the figures reported in Table 1 of the paper.

    (-) In order to reproduce the correct "CF set sizes" reported in Table 1, I had to run the compare.py script with both parameters (file, file2) pointing to the same csv file. This instruction could added to the README file.

---

> ### Public Comment · ~Khanh_Hiep_Tran1 · 2021-08-01
> **Changes made**
>
> Dear Timo,
>
> Thank you for the review. The following changes have been made to improve the artifacts:
>
> (-) The classes and functions are properly documented, but some of the scripts lack code documentation. It would be nice to have a broad overview of the Python modules in RCF/ and NCF/.
>
> ==> More documentation was added and an overview of the directory structures was added in README.
>
> (-) The subdirectories RCF and NCF are not consistent. Maybe it is not feasible due to the different nature of the two approaches, but I see some code fragments (functions) repeated for both approaches (e.g. in accent.py, fia.py). In order to make code maintenance and future re-uses easier, it would be helpful to combine shared functions in single modules, introduce class inheritance, and to harmonize the directory structures.
>
>
> ==> The 'commons' module and inheritance have been added to reduce code repetition as much as possible.
>
> ==> Directories have been restructured to be more consistent.
>
> (-) It is not clear to me which commit was used for the experiments (git tags could be used). I assume, it was the latest commit (the README points to a zipped archive of it). This could have been made more explicit. If the ACCENT framework should be extended in the future, release versioning might also be helpful.
>
> ==> The version used for the experiments has been tagged v1.0 and explicitly mentioned in README.
>
> (-) The data (intermediate results, pretrained models, experimental results, virtual machine) is hosted on Mega. For long-term reproducibility, I'd rather recommend to use a single citable repository, e.g. hosted on Zenodo or figshare.
>
> ==> All data are now hosted on Zenodo.
>
> (-) In order to reproduce the correct "CF set sizes" reported in Table 1, I had to run the compare.py script with both parameters (file, file2) pointing to the same csv file. This instruction could added to the README file.
>
> ==> Code has been modified to accept only the 'file' parameter when the user want to reproduce the set size as in Table 1. Documentation has been added in README.